# Parental Smoking and Under-Five Child Mortality in Southeast Asia: Evidence from Demographic and Health Surveys

**DOI:** 10.3390/ijerph16234756

**Published:** 2019-11-27

**Authors:** Helen Andriani, Septiara Putri, Reynaldi Ikhsan Kosasih, Hsien-Wen Kuo

**Affiliations:** 1Department of Health Policy and Administration, Faculty of Public Health, Universitas Indonesia, 16424 Depok, Indonesia; septiaraputri@ui.ac.id; 2National Committee for Tobacco Control Indonesia, 10350 Jakarta, Indonesia; reynaldi.ikhsan27@gmail.com; 3Institute of Environmental and Occupational Health Sciences, National Yang-Ming University, 112 Taipei, Taiwan; hwkuo@ym.edu.tw

**Keywords:** smoking, children, under-fives mortality, parent, Southeast Asia

## Abstract

Smoking remains the main cause of preventable early death. However, little is known about the association between parental smoking and child mortality in under-fives in developing countries. This study assesses the association between parental smoking status, smoking amount and smoking frequency with child mortality in under-fives in four Southeast Asian countries (Cambodia, Indonesia, Lao People’s Democratic Republic and Timor Leste). We used the Demographic and Health Survey dataset. The information from couples consisting of fathers and mothers (n = 19,301 couples) in the same household were collected. Under-five child mortality was significantly associated with paternal smoking only (odds ratio (OR) = 1.25, 95% confidence interval (CI): 1.14–1.38), maternal smoking only (OR = 2.40, 95% CI: 1.61–3.59) and both parents smoking (OR = 2.60, 2.08–3.26). Paternal, maternal, both parents’ smoking amount and frequency were also assessed. The estimated association decreased after adjusting for covariates but remained highly significant for smoking in both parents, mothers who smoked 1–10 cigarettes/day, when both parents smoked > 20 cigarettes/day, and in mothers who smoked every day. Future behavioural changes and smoking cessation programmes should engage parents as a catalyst for the reduction of child mortality risk in LMICs in the SEA region.

## 1. Introduction

Worldwide, smoking remains the main cause of preventable early death [1]. According to the World Health Organization (WHO) figures, there were six million deaths per year from smoking and 890,000 deaths were from second-hand smoke, and it was responsible for 150,000 deaths in children in 2017 [2]. According to Global Health Observatory data by the WHO, 40% of children globally are exposed to second-hand smoke, which is accountable for more than 600,000 deaths per year, or approximately one in 100 deaths worldwide. Nearly one-third of those who die from cigarette smoke are children under five years of age, two-thirds of which are in Sub-Saharan Africa and Southeast Asia (SEA). Second-hand smoke attributable deaths (per 1000) in children under five years is the highest in LMICs of the SEA region, where children are often exposed to cigarette smoke at home [3]. Compared with Western Europe and the USA (3.7 deaths per 1000 live births and 6.8 deaths per 1,000 live births, respectively), under-five child mortality rates are higher in SEA (40.6/1000) [4]. In Europe and the USA, strict cigarette policies prevents tobacco companies from developing their businesses [2,5].

Tobacco companies have step by step moved the market from high-income countries to low-income countries, where individuals are less educated about the risks of smoking, and anti-tobacco policies are generally frail [6]. Comprehensive, successfully executed and implemented policies of tobacco control are related to noteworthy benefits for child health [7,8], including decreased exposure to second-hand smoke [9,10], fewer preterm births and reduced respiratory and cardiovascular disease [11,12,13,14,15]. Smoking diverts cash from necessities to cigarettes in poor families in developing countries [16] and increases children’s risk of malnutrition [17,18]. Second-hand smoke increases respiratory diseases in children [19]. Perinatal, neonatal and infant death, premature death, low birth weight, preterm birth, small for gestational age, acute respiratory infection, ear problems, congenital anomalies, intensified asthma, Sudden Infant Death Syndrome and slow foetal development and growth are frequent outcomes of exposure to maternal smoking and second-hand smoke exposure during pregnancy [20,21,22,23,24,25,26]. The causal pathways underlying this relationship include intrauterine growth restriction, foetal hypoxia and placental abruption [27]. To date, various examinations have distinguished more than 7000 chemicals and compounds in the inhalation of tobacco smoke, which can cross the placenta and conceivably harm the foetus [28,29], hundreds of which are toxic, including ammonia, polycyclic aromatic hydrocarbons and hydrogen cyanide [30].

The relationship between parental smoking and child health has not been well characterised in developing countries. Previous studies revealed that there were associations between paternal and/or maternal smoking tobacco product use and child death [18,31]. We added critical elements to the existing body of literature by using the most recent and large-scale population-based quantitative data from Demographic and Health Surveys (DHS) and constructing different variables (fathers, mothers and both parents smoking) and taking into account the amount of parental smoking and the frequency of parental smoking with under-five child mortality, neither of which have been presented in previous studies. This may be due to insufficient information on family, economic and environmental factors. The findings of this study will contribute information to the tobacco control policy and programme for policymakers, stakeholders, and decision-makers in SEA countries, especially in LMICs, and support them in developing evidence-based policy related tobacco control by addressing the problems highlighted in this research. Therefore, this study aimed to assess the association between parental smoking status, smoking amount and smoking frequency and under-five child mortality in four SEA countries, namely Cambodia (2014), Indonesia (2017), Lao People’s Democratic Republic (Lao PDR, 2017) and Timor Leste (2016).

## 2. Materials and Methods 

### 2.1. Data Sources

We used the publicly available DHS datasets from four SEA countries. Information was collected from couples consisting of fathers and mothers in the same household. Of the 98,690 households in four countries, 19,301 couples were sampled [32,33,34,35]. The DHS programme, funded by the United States Agency for International Development and other sponsors, and supported by United Nations agencies, gathers nationally representative and comprehensive cross-sectional data from women aged 15 to 49 years, men aged 15 to 54 years and households and children in low- and middle-income countries (LMIC). The DHS sample design utilises two-stage probability samples within the selected enumeration areas and households, using comparable surveys and sampling techniques in all countries [36]. We utilised the latest dataset from every country gathered between 2014 and 2017 in our analysis.

### 2.2. Measurement

Under-five (from birth to 59 months) child mortality was disclosed by parents. The question ‘Have you ever fathered a son or a daughter who was born alive but later died’? (Yes/No) was asked of fathers. For mothers, the question was ‘Have you ever given birth to a boy or girl who was born alive but later died’? (Yes/No) [37]. We recoded the actual number of live births of children who died under the age of five as death ‘1’, and children who did not die in first five years of life as no death ‘0’. Final data reported the number of couples with at least one death of a child under five years of age. Under-five child mortality is the main indicator of child health in developing countries.

Fathers and mothers were classified as ‘smokers’ if they reported currently smoking and ‘non-smokers’ if they did not smoke. A combined variable called ‘parental smoking’ was constructed and categorised into: neither parent (non-smoking fathers and non-smoking mothers); only father (smoking fathers and non-smoking mothers); only mother (non-smoking fathers and smoking mothers); both parents (smoking fathers and smoking mothers). If parental smoking was indicated, parents were further asked about the average number of cigarettes smoked per day. ‘Paternal smoking amount’ or the number of cigarettes fathers smoked per day was classified into four groups as follows: none; 1–10 cigarettes/day; 11–20 cigarettes/day; > 20 cigarettes/day. ‘Maternal smoking amount’ or the number of cigarettes mothers smoked per day was classified into three groups as follows: none; 1–10 cigarettes/day; > 10 cigarettes/day. For ‘smoking amount in both parents’, the three categories were: none; 1–20 cigarettes/day; > 20 cigarettes/day. Both parents’ smoking frequencies were based on the question ‘Do you currently smoke every day, some days or not at all?’ Each of the paternal and maternal smoking frequencies were classified into three groups as follows: not at all; some days; every day.

### 2.3. Potential Covariates

We considered several potential covariates for child mortality, i.e.: residence (urban or rural), mother’s and father’s age separately (<25; 25–39; >39 years), mother’s and father’s education separately (no education; primary; secondary; higher), wealth index (from the 1st quintile or the poorest to the 5th quintile or the richest), mother’s occupation and father’s occupation (did not work; professional; sales and service; worker) and year of survey (year 2016; before or after 2016).

### 2.4. Statistical Analyses

Statistical analyses were performed with SPSS 25.0^®^ (IBM Corporation, Armonk, NY, USA). We utilised country-level data to assess the crude prevalence of parental smoking and under-five child mortality, as well as the proportion of under-five child mortalities as a function of different parental smoking, parental smoking amounts and parental smoking frequency variables. Chi-squared tests were used to test for differences between parental smoking for family, economic and demographic characteristics. Multiple logistic regressions were performed to estimate the odds ratios (ORs) and 95% confidence intervals (CIs) of the unadjusted and adjusted associations between parental smoking, parental smoking amounts and parental smoking frequency with under-five child mortality, adjusting for the covariates. The statistical significance level was 0.05. We used standard weights that were provided by the DHS.

We utilised standard weights given by the DHS and made unique identifiers for primary sampling units (PSUs) and strata in order to pool the data across all four countries. Before pooling the individual datasets, we re-normalised the weights for each survey. In particular, we created three new variables, including country (values from 1 to 4 for the four countries), PSU (renumbering the existing PSU values joined with each country value) and strata (renumbering the existing strata values joined with country values). We calculated the total sum of the weights and each weight was then divided by the total weight. Those weights were multiplied by the country’s sub-population. Analysis was conducted in R software using the complex survey design svyglm, which accounts for two-stage clustering and stratification in the sample design and the unique probabilities of participation and selection. This technique makes the results of different countries comparable. Pooled data for the four countries used the same operational definition to guarantee the homogeneity of each data at the individual level.

### 2.5. Ethical Considerations

The DHS programme collects data periodically for policy development and programme planning, monitoring and evaluation. Respondents read an informed consent statement before each interview is conducted. The statements include voluntary participation, refusal to answer question or termination of participation at any time, and confidentiality of the respondent’s identity and information. Consent prior to a child’s participation had to be provided by a parent or guardian. Procedures and country-specific DHS survey protocols are reviewed and approved by the Inner City Fund International Inc., Fairfax, VA, USA (ICF) Institutional Review Board (IRB). After authorisation to use the dataset was obtained from the DHS, additional ethical review approval was obtained from the IRB in Indonesia, at Universitas Indonesia.

## 3. Results

### 3.1. The Country-Level Demographic Characteristics of the Couples and Under-five Child Mortality

Table 1 presents the demographic characteristics of fathers and mothers at the country-level. The proportion of respondents living in urban areas ranged from 25.9% in Cambodia to 50.0% in Indonesia, and the lowest wealth index ranged from 18.2% in Timor Leste to 25.5% in Lao PDR. The mean age for fathers varied from 34.9 years in Cambodia to 38.9 years in Indonesia and Timor Leste, and from 32.3 years in Cambodia to 35.1 years in Indonesia for mothers. For fathers, no formal education ranged from 1.8% in Indonesia to 27.1% in Timor Leste and from 2.0% in Indonesia to 30.9% in Timor Leste for mothers. The proportion of those not working ranged from 0.3% in Cambodia to 8.9% in Timor Leste for fathers and from 15.7% in Cambodia to 55.9% in Timor Leste for mothers.

Overall, under-five child mortality was highest in Lao PDR (15.6%) and lowest in Indonesia (8.9%) (Table 2). There was a wide variation in the prevalence of parental smoking, parental smoking amounts and parental smoking frequency. The percentages of non-smoking parents, paternal smoking, maternal smoking and both parents smoking were 39.8%, 56.8%, 0.8% and 2.6%, respectively. Paternal smoking varied from a high prevalence of 70.2% in Indonesia to a low prevalence of 38.2% in Cambodia. Maternal smoking ranged from 2.4% in Timor Leste to 0.3% in Indonesia, and both parents smoking was highest in Lao PDR (4.3%) and lowest in Indonesia (1.7%). There were variations in parental smoking amounts. Among fathers who smoked > 20 cigarettes/day, smoking was highest in Indonesia (9.6%) and lowest in Lao PDR (1.5%). Among mothers who smoked >10 cigarettes/day, the highest was Timor Leste (0.6%) and the lowest was Indonesia (0.3%). Overall, 53.5% of fathers smoked every day, varying from a high percentage of 63.1% (Indonesia) to a low percentage of 25.7% (Timor Leste). Overall, 2.2% of mothers smoked every day, varying from a high of 4.4% (Lao PDR) to a low of 0.8% (Indonesia). 

### 3.2. Family, Economic and Demographic Characteristics Associated with Parental Smoking

Significant differences in parental smoking were identified in relation to under-5 child mortality, residence, father’s age, mother’s age, father’s education, mother’s education, father’s occupation, mother’s occupation, wealth index and year of the survey. We observed that 9.6% of the non-smoking parents group were having at least one under-5 child mortality compared to 11.8%, 20.4%, and 21.7% of the paternal, maternal, and both parents’ smoking group, respectively. Compared with non-smoking parents, smoking fathers were more likely to live in an urban area (63.7%), be older than 39 years old (44.0%), have a higher education (11.0%), is a worker (62.1%), lives in poverty (25.6%) or participate in the survey before or after 2016 (92.4%). Compared with non-smoking parents, smoking mothers were more likely to live in an urban area (76.3%), be older than 39 years old (38.1%), have no education (29.0%) and be a worker, (42.9%), lives in poverty (3.6%) or took part in the survey before or after 2016 (69.1%). We found similar results for parents who both smoked (Table 3).

### 3.3. The Association of Parental Smoking, Smoking Amount and Smoking Frequency with Under-Fives Child Mortality

Table 4 shows the associations between parental smoking, smoking amounts and smoking frequency with under-five child mortality for the entire sample. For all countries in the unadjusted model, under-five child mortality was significantly associated with the father’s (OR = 1.25, 95% CI: 1.14–1.38), mother’s (OR = 2.40, 95% CI: 1.61–3.59) or both parents’ smoking (OR = 2.60, 95% CI: 2.08–3.26), paternal and maternal smoking amounts, both parents’ smoking amounts, fathers who smoked every day and maternal smoking frequency. The estimated association decreased after adjusting for the covariates, but remained highly significant for smoking in both parents, mothers who smoked 1–10 cigarettes/day, both parents who smoked > 20 cigarettes/day and mothers who smoked every day.

## 4. Discussion

There was a high overall prevalence of smoking, smoking amounts and smoking frequency among parents in SEA countries, with fathers had a higher prevalence of smoking than mothers or both parents. After controlling for several variables, including household wealth indexes, resident type and parental education level, this study presents strong evidence that with the variation within and between SEA countries, smoking parents increased the risk of under-five child mortality. These results support an Indonesian study that found a significant association between paternal smoking and the risk of infant and under-five child mortality. One of the explanations is that having a smoker in a family diverts the household income from food to tobacco. The amount of cigarette expenditure for low-income families influences spending on nutrition in children, health and education [18]. Maternal smoking prevalence is low; however exposure to maternal smoking probably worsen the harm to children as the tobacco industry increasingly focuses on vulnerable groups and women in poor developing countries [38,39]. The magnitude of the associations of maternal smoking and both parents smoking with under-five child mortality is higher than paternal smoking alone. It is conceivable that children are more exposed to their mothers’ smoking than their fathers’. In most Asian countries, mothers are the primary child carers in the household. Variations in the rate of under-five child mortality across these countries may be because of the utilisation of reproductive health services, socioeconomic and environmental risk factors.

Child death rates and tobacco use in developing countries are consistently high [4,22]. Data from Global Youth Tobacco Survey showed that three out of every five children aged 13–15 years in Indonesia have seen cigarette advertisements at cigarette shops and are exposed to smoking activities both at home and in public places [40]. High exposure to cigarette advertisements and a smoking environment from an early age creates a positive perception of smoking activity in adolescents [41]. The fact that cigarettes are still sold freely makes them accessible to people of all ages. The tobacco industry presents the view of the public, especially to young people, that smoking is a “normal” behaviour. Parental smoking could fundamentally worsen outcomes, and in this way, slow progress in accomplishing the United Nations Sustainable Development Goals (SDGs). Populations in low-income countries are generally unfamiliar with the risks related to tobacco, and the governments of these countries are frequently in need of powerful tobacco control measures. A frequent outcome of this is a high number of tobacco users with restricted information on the risks associated with smoking and second-hand smoke exposure, leading to progressively negative health impact due to second-hand smoke [6,42].

The implementation of the Framework Convention Tobacco Control (FCTC) is a step towards realising this goal. The framework is a binding agreement that aims to protect present and future generations from the health, societal, environmental, and economic consequences of tobacco consumption and exposure to cigarette smoke. This is in the context of protection from all impact on development related to tobacco products, as elaborated in various SDGs objectives [43]. In our study, Indonesia has the lowest mortality rate, yet at the same time, it has the highest paternal smoking. Most fathers in Indonesia work and they may choose to not smoke in the presence of their children. In addition, maternal and child health services (monitoring child health status and nutrition) and programs (compulsory immunisation, nutrition education, exclusive breastfeeding, weaning) have been improved, supported by the implementation of Indonesia’s National Health Insurance in 2014, which opened up more extensive opportunities for more accessible healthcare [44]. Indonesia is the only SEA country that has not accessed FCTC, making it a paradise for the cigarette industry, where public health is damaged and the competitiveness of human resources is weakened, creating further disparities between Indonesia and other countries. Cambodia has reported negligible NRT and/or some additional cessation services with no costs covered [2,5]. Timor Leste has no cessation strategies or programme for the treatment of tobacco reliance and Lao PDR has reported on tobacco-related costs and the benefits of the cessation of tobacco use and tobacco-free lifestyles [2,45]. Ratification or accession of the FCTC ensures that a country is moving in the right direction to achieve world development targets. Integrating tobacco cessation interventions with maternal health care services in developing countries would, therefore, reduce the adverse health impacts of second-hand smoke. 

Our study potentially benefits policymakers by providing evidence to support tobacco control by LMIC, where these issues have been relatively neglected. Policymakers in LMIC ought to coordinate codes of conduct for cross-country cigarette promotions, sponsorships and adverts, particularly in SEA. The Ministry of Health ought to execute or amend laws and regulations to discourage smoking in smoke-free zones and around pregnant women, increase tobacco taxes and control media transmission of tobacco advertising. This would fortify the FCTC, effectively advance behavioural changes and support parental smoking cessation programmes through educational campaigns and media coverage of the adverse health effects of tobacco smoke.

A major strength of our study included the large representative DHS sample of the SEA population. This applies standard procedures and measurement units to the selection of primary sampling units, household strata and respondents, which therefore increases the generalizability of our findings. We utilised multi-country data to demonstrate the association between smoking, the amount and frequency of exposure to second-hand tobacco smoke and under-five child mortality after adjusting for several covariates. The indicators we used for second-hand tobacco smoke (smoking amount and frequency) enabled us to adequately explain the true exposure amount or a true dose-response relationship. Finally, the use of the most recent data across a short range (2014 to 2017) minimised potential bias associated with time-effects.

There were several limitations to our study. First, the data used in this study was pulled from a retrospective cross-sectional survey. We, therefore, could not establish temporal relationships, and our findings need to be interpreted with caution. Future investigations should consider a prospective study to evaluate the cause and effect relationships of second-hand smoke and child mortality. Second, our study is limited because other biological factors contribute to under-five child mortality (such as low birth weight and prematurity), other environment health risks (indoor or outdoor non-tobacco smoke exposure, number of household members who smoke and ambient air pollution), health-related factors and children’s cause of death (such as unavailability, inaccessibility and low utilization of maternal and child healthcare services, as well as a historical comprehensive health evaluation of the children). Third, parental smoking and child mortality-related information was based on self-reported data, which might not reflect the true prevalence and may have been influenced by recall bias. This raises the possibility of under-reporting. Future research utilising biomarkers of smoking exposure (e.g., the levels of cotinine in urine or hair) could improve the accuracy of self-reported DHS data. Fourth, childhood mortality rates were based on a historical data and, therefore, not concurrent with the smoking status at the time of the survey. The death of a child, however, might alter the number of cigarettes smoked.

## 5. Conclusions

Parental smoking was positively associated with under-five child mortality in the countries we studied. Mothers’ and both parents’ smoking had a more dominant influence on the risk of under-five mortality than did the exposure to paternal smoking. Policymakers in LMICs in the SEA region should aggressively implement anti-smoking policy reform and law amendments in public places, workplaces, and families. More public health promotion through behavioural changes and parental smoking cessation programmes to educate parents on the health risk of secondhand tobacco smoke is needed. This includes not smoking around pregnant women, increasing tobacco taxes and controlling media transmission of tobacco advertising to fortify the FCTC.

## Figures and Tables

**Table 1 ijerph-16-04756-t001:** Demographic characteristics of fathers and mothers.

Country, year	Cambodia, 2014	Indonesia, 2017	Lao PDR, 2017	Timor Leste, 2016
Total # couples	3051	8831	5359	1972
Urban resident	25.90%	50.00%	27.40%	28.60%
Lowest wealth quintile	18.40%	22.20%	25.50%	18.20%
Mean age in years (father)	34.9	38.9	37.2	38.9
Mean age in years (mother)	32.3	35.1	33.9	34
No formal education (father)	8.20%	1.80%	10.60%	27.10%
No formal education (mother)	13.80%	2.00%	24.40%	30.90%
Not working (father)	0.30%	0.50%	n.a.	8.90%
Not working (mother)	15.70%	36.70%	n.a.	55.90%

Note: n.a. = not available.

**Table 2 ijerph-16-04756-t002:** Prevalence of smoking among parents who smoked and under-5 child mortality, percent (95% CI).

**Country, year**	**Parental smoking**						**Under-5 child mortality**
	**Only father**		**Only mother**		**Both parents**		
Cambodia, 2014	38.2 (36.4, 39.9)		0.9 (0.5, 1.2)		2.2 (1.6, 2.7)		11.1 (9.9, 12.2)
Indonesia, 2017	70.2 (69.2, 71.1)		0.3 (0.2, 0.4)		1.7 (1.4, 1.9)		8.9 (8.3, 9.4)
Lao PDR, 2017	50.6 (49.2, 51.9)		1.0 (0.7, 1.2)		4.3 (3.7, 4.8)		15.6 (14.6, 16.5)
Timor Leste, 2016	42.4 (40.2, 44.5)		2.4 (0.2, 0.3)		2.9 (2.1, 3.6)		10.2 (8.8, 11.5)
Overall	56.8 (56.1, 57.5)		0.8 (0.7, 0.9)		2.6 (2.4, 2.8)		11.3 (10.9, 11.8)
**Country, year**	**Father**				**Mother**		**Under-5 child mortality**
	**1–10 cigarettes/day**	**11–20 cigarettes/day**		**> 20 cigarettes/day**	**1–10 cigarettes/day**	**> 10 cigarettes/day**	
Cambodia, 2014	17.9 (16.5, 19.2)	19.4 (17.9, 20.8)		3.2 (2.5, 3.8)	2.6 (2.0, 3.1)	0.5 (0.2, 0.7)	11.1 (9.9, 12.2)
Indonesia, 2017	22.2 (21.3, 23.0)	40 (38.9, 41.0)		9.6 (8.9, 10.2)	1.7 (1.4, 1.9)	0.3 (0.18, 0.4)	8.9 (8.3, 9.4)
Lao PDR, 2017	33.1 (31.8, 34.3)	19.6 (18.5, 20.6)		1.5 (1.1, 1.8)	4.9 (4.3, 5.4)	0.4 (0.2, 0.5)	15.6 (14.6, 16.5)
Timor Leste, 2016	27.7 (25.7, 29.6)	11.3 (9.9, 12.6)		6.4 (5.3, 7.4)	4.8 (3.8, 5.7)	0.6 (0.2, 0.9)	10.2 (8.8, 11.5)
Overall	32.1 (31.4, 32.8)	35.9 (35.2, 36.6)		7.7 (7.3, 8.1)	4.0 (3.7, 4.3)	0.5 (0.4, 0.6)	11.3 (10.9, 11.8)
**Country, year**	**Father**				**Mother**		**Under-5 child mortality**
	**Some days**		**Every day**		**Some days**	**Every day**	
Cambodia, 2014	n.a.		n.a.		n.a.	n.a.	11.1 (9.9, 12.2)
Indonesia, 2017	8.8 (7.7, 9.8)		63.1 (60.9, 65.2)		1.1 (0.8, 1.3)	0.8 (6.1, 9.8)	8.9 (8.3, 9.4)
Lao PDR, 2017	6.2 (5.6, 6.7)		48.0 (46.2, 49.7)		0.7 (0.3, 1.0)	4.4 (3.8, 4.9)	15.6 (14.6, 16.5)
Timor Leste, 2016	19.7 (18.6, 20.7)		25.7 (24.7, 26.6)		3.5 (2.8, 4.1)	1.8 (1.2, 2.3)	10.2 (8.8, 11.5)
Overall	9.6 (9.2, 10.0)		53.3 (52.6, 54.0)		1.3 (1.1, 1.5)	2.2 (2.0, 2.4)	11.3 (10.9, 11.8)

Note: n.a. = not available.

**Table 3 ijerph-16-04756-t003:** Participants and their family characteristics across categories of parental smoking.

Characteristic	Neither Parent n (%)	Only Father n (%)	Only Mother n (%)	Both Parents n (%)	*p*
Under-5 child mortality					<0.001
No death	6945 (90.4)	9666 (88.2)	121 (79.6)	396 (78.3)	
At least one death	741 (9.6)	1291 (11.8)	31 (20.4)	110 (21.7)	
Residence					<0.001
Rural	3123 (40.6)	3982 (36.3)	36 (23.7)	130 (25.7)	
Urban	4563 (59.4)	6975 (63.7)	116 (76.3)	376 (74.3)	
Father’s age					<0.001
<25	434 (5.6)	494 (4.5)	10 (6.6)	11 (2.2)	
25–39	3962 (51.5)	5639 (51.5)	58 (38.2)	228 (45.1)	
>39	3290 (42.8)	4824 (44.0)	84 (55.3)	267 (52.8)	
Mother’s age					<0.001
<25	1033 (13.4)	1297 (11.8)	14 (9.2)	23 (4.5)	
25–39	4583 (59.6)	6499 (59.3)	290 (57.3)	290 (57.3)	
>39	2070 (26.9)	3161 (28.8)	193 (38.1)	193 (38.1)	
Father’s education					<0.001
No education	370 (4.9)	513 (4.8)	18 (12.9)	48 (10.6)	
Primary	2253 (30.0)	4204 (39.6)	63 (45.0)	246 (54.2)	
Secondary	3159 (42.1)	4738 (44.6)	42 (30.0)	137 (30.2)	
Higher	1717 (22.9)	1174 (11.0)	17 (12.1)	23 (5.1)	
Mother’s education					<0.001
No education	480 (6.7)	621 (6.1)	36 (29.0)	73 (18.3)	
Primary	2592 (36.0)	4216 (41.2)	49 (39.5)	212 (53.0)	
Secondary	2976 (41.3)	4375 (42.7)	30 (24.2)	101 (25.3)	
Higher	1156 (16.0)	1026 (10.0)	9 (7.3)	14 (3.5)	
Father’s occupation					<0.001
Did not work	143 (2.7)	84 (1.0)	3 (3.1)	1 (0.4)	
Professional	1097 (20.8)	946 (11.5)	18 (18.4)	29 (10.6)	
Sales and Service	1141 (21.6)	2077 (25.3)	9 (9.2)	47 (17.2)	
Worker	2895 (54.9)	5093 (62.1)	68 (69.4)	196 (71.8)	
Mother’s occupation					<0.001
Did not work	1719 (32.6)	2991 (36.5)	29 (29.6)	76 (27.8)	
Professional	711 (13.5)	722 (8.8)	14 (14.3)	13 (4.8)	
Sales and Service	1372 (26.0)	2034 (24.8)	13 (13.3)	67 (24.5)	
Worker	1476 (28.0)	2449 (29.9)	42 (42.9)	117 (42.9)	
Wealth Index					<0.001
Q1 (poorest)	1201 (15.6)	2800 (25.6)	51 (33.6)	210 (41.5)	
Q2	1377 (17.9)	2495 (22.8)	36 (23.7)	121 (23.9)	
Q3	1465 (19.1)	2186 (20.0)	23 (15.1)	81 (16.0)	
Q4	1653 (21.5)	1956 (17.9)	24 (15.8)	60 (11.9)	
Q5 (richest)	1990 (25.9)	1520 (13.9)	18 (11.8)	34 (6.7)	
Year of survey					<0.001
Year 2016	1030 (13.4)	837 (7.6)	47 (30.9)	58 (11.5)	
Before or after 2016	6656 (86.6)	10120 (92.4)	105 (69.1)	448 (88.5)	

**Table 4 ijerph-16-04756-t004:** Associations of parental smoking, smoking amount and smoking frequency with under-5 child mortality in four South East Asian countries.

Parental Smoking	Under-5 Child Mortality			
	OR (95% CI)	*p*	OR (95% CI)	*p*
Parental smoking				
Neither parent	1		1	
Only father	1.25 (1.14, 1.38)	<0.001	1.16 (1.04, 1.29) ^a^	0.007
Only mother	2.40 (1.61, 3.59)	<0.001	2.19 (1.40, 3.41) ^a^	0.001
Both parents	2.60 (2.08, 3.26)	<0.001	1.91 (1.46, 2.50) ^a^	<0.001
Paternal smoking amount				
None	1		1	
1–10 cigarettes/day	1.68 (1.46, 1.95)	<0.001	1.28 (1.10, 1.50) ^b^	0.001
11–20 cigarettes/day	1.45 (1.26, 1.68)	<0.001	1.18 (1.02, 1.37) ^b^	0.031
>20 cigarettes/day	1.56 (1.26, 193)	<0.001	1.34 (1.08, 1.66) ^b^	0.009
Maternal smoking amount				
None	1		1	
1–10 cigarettes/day	2.08 (1.70, 2.56)	<0.001	1.58 (1.23, 2.03) ^c^	<0.001
>10 cigarettes/day	3.12 (1.88, 5.19)	<0.001	1.93 (1,08, 3.46) ^c^	0.028
Both parents’ smoking amount				
None	1		1	
1–20 cigarettes/day	1.66 (1.41, 1.95)	<0.001	1.19 (1.00, 1.41) ^a^	0.050
>20 cigarettes/day	2.34 (1.90, 2.89)	<0.001	1.56 (1.24, 1.95) ^a^	<0.001
Paternal smoking frequency				
Not at all	1		1	
Some days	1.24 (1.11, 1.37)	0.711	0.91 (0.75, 1.10) ^b^	0.328
Every day	1.24 (1.11, 1.37)	<0.001	1.08 (0.97, 1.21) ^b^	0.179
Maternal smoking frequency				
Not at all	1		1	
Some days	1.67 (1.16, 2.40)	0.005	1.56 (1.05, 2.30) ^c^	0.026
Every day	2.41 (1.87, 3.11)	<0.001	2.02 (1.48, 2.76) ^c^	<0.001

^a^ Adjusting for residence, mother’s education, father’s education, wealth index; ^b^ Adjusting for residence, father’s education, wealth index; ^c^ Adjusting for residence, mother’s education, wealth index.

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
