# Peer review of "Parental Smoking and Under-Five Child Mortality in Southeast Asia: Evidence from Demographic and Health Surveys"

_ijerph, 2019, doi:10.3390/ijerph16234756_

Round 1

Reviewer 1 Report

This paper use the most recent and large-scale population-based quantitative data from DHS and the results will expand the generalizability regarding population representativeness of this topic. So I think it has good value to knowledge in this research field and is acceptable for publication.

Reviewer 2 Report

I read with great attention this manuscript titled by Parental Smoking and Under-Five Child Mortality in 2 Southeast Asia: Evidence from Demographic and 3 Health Surveys. The manuscript is interesting and well conducted.
Introduction to the study present shortly current knowledge related to the undertaken study and it is base of properly selected scientific literature.
The study is based in simple, efficient, and correct methodology that provides some interesting results.

It is worth emphasizing well thought out and well balanced discussion.

The manuscript is interesting not only in terms of research, but public health and clinical use.

This manuscript is a resubmission of an earlier submission. The following is a list of the peer review reports and author responses from that submission.

Round 1

Reviewer 1 Report

This manuscript presented an interesting topic about smoking and under-five child mortality in SEA. 

Since DHS sample design utilizes two-stage probability samples, when describing the prevalence, it may be more appropriate to use specific statistical method for survey data, taking into account the complex survey. There might be heterogeneity for the association between smoking and mortality among different countries, so a multilevel model might be more appropriate. In the current analysis, combined variables for parental smoking status are derived. But this may be inappropriate since the mother's smoking behavior could affect the mortality to a different extent than the father's. Separate variables for mothers and fathers can be created and an interaction term can be used to see if smoking for both mother and father can have a greater impact.

Author Response

Response to Reviewer 1 Comments

Point 1: This manuscript presented an interesting topic about smoking and under-five child mortality in SEA.

Response 1: We thank and appreciate the constructive comments of the reviewer that will significantly improve the manuscript. We have addressed each of the concerns as outlined below. We also have addressed some major issues in the revised version of the manuscript based on your comments and used the "Track Changes" function in Microsoft Word, so that they are easily visible to the reviewer. 

Point 2: Since DHS sample design utilizes two-stage probability samples, when describing the prevalence, it may be more appropriate to use specific statistical method for survey data, taking into account the complex survey. There might be heterogeneity for the association between smoking and mortality among different countries, so a multilevel model might be more appropriate.

Response 2: We utilized standard weights that were given by the DHS. We made unique identifiers for primary sampling units (PSUs) and strata to pool the data across all four countries and re-normalized the weights for each survey before pooling the individual country datasets. In particular, we created new variables including country (values from 1 to 4 for the four countries), PSU (renumbered the existing PSU values joined with country value to make all the PSUs unique in the pooled dataset) and strata (renumbered the existing strata values joined with country value to make them unique). Weights were adjusted by calculating the total sum of weights, and individual weight divided by the total weights, and afterward multiplied those weights by the country’s sub-population.

To account for the two-stage cluster sampling using sampling weights given by DHS, analysis was done in R software using the complex survey analysis function svyglm, which account for clustering and stratification in the sampling design and unique probabilities of participation and selection. This technique makes results among different countries comparable.

Pooled data in four countries (Table 3 and Table 4) used the same operational definition to guarantee the homogeneity of each data at the individual level.

Point 3: In the current analysis, combined variables for parental smoking status are derived. But this may be inappropriate since the mother's smoking behavior could affect the mortality to a different extent than the father's. Separate variables for mothers and fathers can be created and an interaction term can be used to see if smoking for both mother and father can have a greater impact.

Response 3: We have separated smoking amount and smoking frequency variables for mothers and fathers, however for parental smoking, we used a combined variable (4 groups of parental smoking status) instead of separated variables (paternal smoking and maternal smoking) to help us getting more clear results regarding the association between parental smoking and under-5 child mortality whether the smoking exposure is from neither parent, one parent (only father or only mother), and both parents. If we conduct further analysis with separated variables for fathers (smokers and non-smokers) and mothers (smokers and non-smokers), that would be more challenging since we could not specify whether the spouse is a smoker or not.

We hope our revision meet your approval. Again, we thank the reviewer and appreciate the opportunity to revise our work for consideration for publication in IJERPH.

Reviewer 2 Report

The topic covered in this paper is most important and it is gratifying to see researchers making use of existing datasets (information gathered at high cost) instead of starting new data collections. This is an excellent initiative and the authors are to be congratulated for this. 

A basic problem with the paper is that I am not clear on who the audience is. And that is fundamental for the methods and information to be used to advance arguments in the introduction, discussion and conclusion sections of the paper.

Is it a paper to the world community or to all developing countries about the impact of smoking on child health? And therefore is the aim to enthuse policymakers to accelerate tobacco control and parents everywhere (especially in the developing world) to quit smoking for the benefit of their children’s health. In lines 59-60, the authors state that “The relationship between parental smoking and child health has not been well documented in developing countries.” And again in lines 65-68, the authors make a case that “By using the most recent and large-scale quantitative data from Demographic and Health Surveys (DHS) conducted between 2014 and 2017, the findings of this study will contribute information to the tobacco control policy and programme for decision makers in developing countries”

If that is the case, the authors of the paper have to explain why they have analysed data from only four SEA countries when DHS has been conducted in many developing countries. And why only in the window 2014-2017. There may be very valid reasons for both but the authors need to explain why their choice of the 4 countries and the time frame selected provides the optimal information for policy makers in developing countries everywhere.

Or is it a paper aimed primarily at SEA countries (and maybe in particular to the four selected countries). If aimed at SEA countries specifically, then what is the rationale for why the four selected countries only. The DHS website notes that DHS has also been carried out in the following Southeast Asian countries: Philippines, Thailand and Viet Nam. (https://www.dhsprogram.com/Where-We-Work/Country-List.cfm), and again why only 2014-2017.

Or is the focus of the paper intended to be on the the four selected countries only – see emphasis of last sentence of the introduction.

There are also some inconsistencies in the Introduction. The authors claim that the study is needed because of a lack of knowledge. But yet, lines 35-38 presented information on the impact of SHS in sub-Saharan Africa and SEA, and again in lines 60 and 61 after stating that “previous studies revealed that there were associations between paternal and/or maternal smoking and child death”. They cite article 18 - a study in on paternal smoking and under-five child mortality in Indonesia – and article 31 – a study using DHS data in South and South East Asia exploring links between parental smoking and child health. However It is confusing that almost in the same breath, the authors go on to say in lines 61-64 “ a population based study comprehensively testing the association of fathers, mothers and both parents smoking … with under-five child mortality in SEA countries has yet to be conducted”. Is it that by comprehensively testing, they meant to collapse the data from several surveys to obtain an overall estimate. If yes, then the question remains as to why only the four selected countries.

Methods and results

The authors provide a good description of how they have derived the four categories of smoking (Non-smoker, Paternal only, Maternal only and Both parents smoking). A limitation of the data as presented is that these are most probably linked to current smoking status at the time of the survey. The death of the children occurred in the past and therefore not concurrent with smoking status at the time. This may not be a fatal problem however as most people start smoking in their adolescence and given the addictiveness of the product, maybe smoking status at the time of the survey may indeed reflect smoking status at the time of death of the children. Quantity smoked however may indeed have changed. Maybe the death of a child altered the number of cigarettes smoked (it could in both directions). Indeed whilst the analysis by smoker status yields likely results, the analysis by number of cigarettes smoked provide results that appear contradictory. For example, paternal smoking 11-20 cigarettes yield ORs that are better than for 20+. Maybe if the ORs are presented to one decimal place rather than 2

There is not much information on how the death history is constructed. It may be useful to present in Table 3 the number of child deaths by smoking characteristic of parent. And what is the unit of analysis. Was it actual number of children born alive who died before age 5, and actual number of children born alive and who did not die in first five years of life cross tabulated by say paternal smoking status’ Or was it the number of families with a history of at least one under-five child death.

A review of the child death rates by country presented in table 2 is interesting. Indonesia, Cambodia and Lao are all statistically different from each other. Timor-Leste has rates that are consistent with Indonesia and Cambodia. The authors go on to collapse the data across the first three countries without any commentary. Also why is it that Indonesia has the lowest mortality rate at the same time that it has the highest paternal smoking. Is it because they have the lowest maternal smoking rate is lower in that country.

Discussion and conclusion

I have read the discussion and conclusion but there is no point reviewing these in detail until the authors consider the suggested actions presented above.

The topic covered in this paper is most important and it is gratifying to see researchers making use of existing datasets (information gathered at high cost) instead of starting new data collections. This is an excellent initiative and the authors are to be congratulated for this. 

A basic problem with the paper is that I am not clear on who the audience is. And that is fundamental for the methods and information to be used to advance arguments in the introduction, discussion and conclusion sections of the paper.

Is it a paper to the world community or to all developing countries about the impact of smoking on child health? And therefore is the aim to enthuse policymakers to accelerate tobacco control and parents everywhere (especially in the developing world) to quit smoking for the benefit of their children’s health. In lines 59-60, the authors state that “The relationship between parental smoking and child health has not been well documented in developing countries.” And again in lines 65-68, the authors make a case that “By using the most recent and large-scale quantitative data from Demographic and Health Surveys (DHS) conducted between 2014 and 2017, the findings of this study will contribute information to the tobacco control policy and programme for decision makers in developing countries”

If that is the case, the authors of the paper have to explain why they have analysed data from only four SEA countries when DHS has been conducted in many developing countries. And why only in the window 2014-2017. There may be very valid reasons for both but the authors need to explain why their choice of the 4 countries and the time frame selected provides the optimal information for policy makers in developing countries everywhere.

Or is it a paper aimed primarily at SEA countries (and maybe in particular to the four selected countries). If aimed at SEA countries specifically, then what is the rationale for why the four selected countries only. The DHS website notes that DHS has also been carried out in the following Southeast Asian countries: Philippines, Thailand and Viet Nam. (https://www.dhsprogram.com/Where-We-Work/Country-List.cfm), and again why only 2014-2017.

Or is the focus of the paper intended to be on the the four selected countries only – see emphasis of last sentence of the introduction.

There are also some inconsistencies in the Introduction. The authors claim that the study is needed because of a lack of knowledge. But yet, lines 35-38 presented information on the impact of SHS in sub-Saharan Africa and SEA, and again in lines 60 and 61 after stating that “previous studies revealed that there were associations between paternal and/or maternal smoking and child death”. They cite article 18 - a study in on paternal smoking and under-five child mortality in Indonesia – and article 31 – a study using DHS data in South and South East Asia exploring links between parental smoking and child health. However It is confusing that almost in the same breath, the authors go on to say in lines 61-64 “ a population based study comprehensively testing the association of fathers, mothers and both parents smoking … with under-five child mortality in SEA countries has yet to be conducted”. Is it that by comprehensively testing, they meant to collapse the data from several surveys to obtain an overall estimate. If yes, then the question remains as to why only the four selected countries.

Methods and results

The authors provide a good description of how they have derived the four categories of smoking (Non-smoker, Paternal only, Maternal only and Both parents smoking). A limitation of the data as presented is that these are most probably linked to current smoking status at the time of the survey. The death of the children occurred in the past and therefore not concurrent with smoking status at the time. This may not be a fatal problem however as most people start smoking in their adolescence and given the addictiveness of the product, maybe smoking status at the time of the survey may indeed reflect smoking status at the time of death of the children. Quantity smoked however may indeed have changed. Maybe the death of a child altered the number of cigarettes smoked (it could in both directions). Indeed whilst the analysis by smoker status yields likely results, the analysis by number of cigarettes smoked provide results that appear contradictory. For example, paternal smoking 11-20 cigarettes yield ORs that are better than for 20+. Maybe if the ORs are presented to one decimal place rather than 2

There is not much information on how the death history is constructed. It may be useful to present in Table 3 the number of child deaths by smoking characteristic of parent. And what is the unit of analysis. Was it actual number of children born alive who died before age 5, and actual number of children born alive and who did not die in first five years of life cross tabulated by say paternal smoking status’ Or was it the number of families with a history of at least one under-five child death.

A review of the child death rates by country presented in table 2 is interesting. Indonesia, Cambodia and Lao are all statistically different from each other. Timor-Leste has rates that are consistent with Indonesia and Cambodia. The authors go on to collapse the data across the first three countries without any commentary. Also why is it that Indonesia has the lowest mortality rate at the same time that it has the highest paternal smoking. Is it because they have the lowest maternal smoking rate is lower in that country.

Discussion and conclusion

I have read the discussion and conclusion but there is no point reviewing these in detail until the authors consider the suggested actions presented above.

The topic covered in this paper is most important and it is gratifying to see researchers making use of existing datasets (information gathered at high cost) instead of starting new data collections. This is an excellent initiative and the authors are to be congratulated for this. 

A basic problem with the paper is that I am not clear on who the audience is. And that is fundamental for the methods and information to be used to advance arguments in the introduction, discussion and conclusion sections of the paper.

Is it a paper to the world community or to all developing countries about the impact of smoking on child health? And therefore is the aim to enthuse policymakers to accelerate tobacco control and parents everywhere (especially in the developing world) to quit smoking for the benefit of their children’s health. In lines 59-60, the authors state that “The relationship between parental smoking and child health has not been well documented in developing countries.” And again in lines 65-68, the authors make a case that “By using the most recent and large-scale quantitative data from Demographic and Health Surveys (DHS) conducted between 2014 and 2017, the findings of this study will contribute information to the tobacco control policy and programme for decision makers in developing countries”

If that is the case, the authors of the paper have to explain why they have analysed data from only four SEA countries when DHS has been conducted in many developing countries. And why only in the window 2014-2017. There may be very valid reasons for both but the authors need to explain why their choice of the 4 countries and the time frame selected provides the optimal information for policy makers in developing countries everywhere.

Or is it a paper aimed primarily at SEA countries (and maybe in particular to the four selected countries). If aimed at SEA countries specifically, then what is the rationale for why the four selected countries only. The DHS website notes that DHS has also been carried out in the following Southeast Asian countries: Philippines, Thailand and Viet Nam. (https://www.dhsprogram.com/Where-We-Work/Country-List.cfm), and again why only 2014-2017.

Or is the focus of the paper intended to be on the the four selected countries only – see emphasis of last sentence of the introduction.

There are also some inconsistencies in the Introduction. The authors claim that the study is needed because of a lack of knowledge. But yet, lines 35-38 presented information on the impact of SHS in sub-Saharan Africa and SEA, and again in lines 60 and 61 after stating that “previous studies revealed that there were associations between paternal and/or maternal smoking and child death”. They cite article 18 - a study in on paternal smoking and under-five child mortality in Indonesia – and article 31 – a study using DHS data in South and South East Asia exploring links between parental smoking and child health. However It is confusing that almost in the same breath, the authors go on to say in lines 61-64 “ a population based study comprehensively testing the association of fathers, mothers and both parents smoking … with under-five child mortality in SEA countries has yet to be conducted”. Is it that by comprehensively testing, they meant to collapse the data from several surveys to obtain an overall estimate. If yes, then the question remains as to why only the four selected countries.

Methods and results

The authors provide a good description of how they have derived the four categories of smoking (Non-smoker, Paternal only, Maternal only and Both parents smoking). A limitation of the data as presented is that these are most probably linked to current smoking status at the time of the survey. The death of the children occurred in the past and therefore not concurrent with smoking status at the time. This may not be a fatal problem however as most people start smoking in their adolescence and given the addictiveness of the product, maybe smoking status at the time of the survey may indeed reflect smoking status at the time of death of the children. Quantity smoked however may indeed have changed. Maybe the death of a child altered the number of cigarettes smoked (it could in both directions). Indeed whilst the analysis by smoker status yields likely results, the analysis by number of cigarettes smoked provide results that appear contradictory. For example, paternal smoking 11-20 cigarettes yield ORs that are better than for 20+. Maybe if the ORs are presented to one decimal place rather than 2

There is not much information on how the death history is constructed. It may be useful to present in Table 3 the number of child deaths by smoking characteristic of parent. And what is the unit of analysis. Was it actual number of children born alive who died before age 5, and actual number of children born alive and who did not die in first five years of life cross tabulated by say paternal smoking status’ Or was it the number of families with a history of at least one under-five child death.

A review of the child death rates by country presented in table 2 is interesting. Indonesia, Cambodia and Lao are all statistically different from each other. Timor-Leste has rates that are consistent with Indonesia and Cambodia. The authors go on to collapse the data across the first three countries without any commentary. Also why is it that Indonesia has the lowest mortality rate at the same time that it has the highest paternal smoking. Is it because they have the lowest maternal smoking rate is lower in that country.

Discussion and conclusion

I have read the discussion and conclusion but there is no point reviewing these in detail until the authors consider the suggested actions presented above.

Author Response

Response to Reviewer 2 Comments

Point 1: Extensive editing of English language and style required 

Response 1: Our manuscript has been entirely reviewed by a native English editor and written in correct scientific English. 

Point 2: The topic covered in this paper is most important and it is gratifying to see researchers making use of existing datasets (information gathered at high cost) instead of starting new data collections. This is an excellent initiative and the authors are to be congratulated for this.

Response 2: Thank you for your thorough review, salient observations, and for helpful comments that will significantly improve the manuscript. We have addressed some major issues in the revised version of the manuscript based on your comments and used the "Track Changes" function in Microsoft Word, so that they are easily visible to the reviewer. 

Point 3: A basic problem with the paper is that I am not clear on who the audience is. And that is fundamental for the methods and information to be used to advance arguments in the introduction, discussion and conclusion sections of the paper.

Response 3: This paper aims to provide evidence for scholars, researchers, and stakeholders interested in the topic of under-5 years old child death and or tobacco control issue. Specifically, we hope that this finding can reach prominent policymakers and stakeholders in Southeast Asia (SEA) countries with Lower Middle-Income Countries (LMICs) classification, to support them developing evidence-based policy related tobacco control and addressing the problems found in this research.

Point 4: Is it a paper to the world community or to all developing countries about the impact of smoking on child health? And therefore is the aim to enthuse policymakers to accelerate tobacco control and parents everywhere (especially in the developing world) to quit smoking for the benefit of their children’s health. In lines 59-60, the authors state that “The relationship between parental smoking and child health has not been well documented in developing countries.” And again in lines 65-68, the authors make a case that “By using the most recent and large-scale quantitative data from Demographic and Health Surveys (DHS) conducted between 2014 and 2017, the findings of this study will contribute information to the tobacco control policy and programme for decision makers in developing countries. If that is the case, the authors of the paper have to explain why they have analysed data from only four SEA countries when DHS has been conducted in many developing countries. And why only in the window 2014-2017. There may be very valid reasons for both but the authors need to explain why their choice of the 4 countries and the time frame selected provides the optimal information for policy makers in developing countries everywhere.

Response 4: We did not intend to limit the time frame in the window 2014-2017; instead we selected SEA countries classified as LMICs for analysis using the most recent dataset from each country, i.e., Cambodia in 2014, Timor Leste in 2016, Indonesia in 2017, and Lao PDR in 2017. Therefore the time frame used in this study was from 2014 to 2017.

We are focusing our study on SEA region because based on Global Health Observatory data repository, second-hand smoke attributable deaths (‘000) in children under 5 years was the largest in LMICs of the SEA region, also considering the higher rates of under-five child mortality, the lack of education on health risks due to smoking, as well as the weaknesses of anti-tobacco policies in SEA region compared to others, hence this study allowed us to give applicative tobacco-related policy recommendations that might be relevant in SEA countries, especially for LMICs. We have adjusted the text to be more explicit in the Introduction section of the revised manuscript.

Point 5: Or is it a paper aimed primarily at SEA countries (and maybe in particular to the four selected countries). If aimed at SEA countries specifically, then what is the rationale for why the four selected countries only. The DHS website notes that DHS has also been carried out in the following Southeast Asian countries: Philippines, Thailand and Viet Nam. (https://www.dhsprogram.com/Where-We-Work/Country-List.cfm), and again why only 2014-2017. Or is the focus of the paper intended to be on the the four selected countries only – see emphasis of last sentence of the introduction.

Response 5: Our study aimed for SEA region, classified as LMICs. Thailand (DHS 1987) was not included in our study since it is classified as Upper Middle-Income country. We did not include Sri Lanka (DHS 2007) since the DHS data is not in the public domain. Lastly, we did not include Vietnam (AIS 2005) and Philippines (DHS 2017) because there was no tobacco use data for both countries.

Point 6: There are also some inconsistencies in the Introduction. The authors claim that the study is needed because of a lack of knowledge. But yet, lines 35-38 presented information on the impact of SHS in sub-Saharan Africa and SEA, and again in lines 60 and 61 after stating that “previous studies revealed that there were associations between paternal and/or maternal smoking and child death”. They cite article 18 - a study in on paternal smoking and under-five child mortality in Indonesia – and article 31 – a study using DHS data in South and South East Asia exploring links between parental smoking and child health. However It is confusing that almost in the same breath, the authors go on to say in lines 61-64 “ a population based study comprehensively testing the association of fathers, mothers and both parents smoking … with under-five child mortality in SEA countries has yet to be conducted”. Is it that by comprehensively testing, they meant to collapse the data from several surveys to obtain an overall estimate. If yes, then the question remains as to why only the four selected countries.

Response 6: Previous studies in article 18 reported paternal smoking and under-five child mortality in Indonesia from 2000 to 2003 and article 31 using DHS data reported the association between parental tobacco products use and child death by not using the most recent DHS dataset available, i.e., Indonesia (2012) and Timor-Leste (2010). By utilizing the most recent data in the four countries, i.e. Cambodia (2014), Timor Leste (2016), Indonesia (2017), and Lao PDR (2017), we added essential aspects of the existing body of literature by constructing different variables (paternal smoking, maternal smoking, and both parents’ smoking) and taking into account paternal and maternal smoking amount and frequency that are not yet available in previous studies. It is our sincere hope that this manuscript provides more insightful information or evidence in terms of under-5 child mortality and parental smoking as well as the necessary significance and implications for tobacco control policy especially among LMICs in SEA region.

Point 7: Methods and results.

The authors provide a good description of how they have derived the four categories of smoking (Non-smoker, Paternal only, Maternal only and Both parents smoking). A limitation of the data as presented is that these are most probably linked to current smoking status at the time of the survey. The death of the children occurred in the past and therefore not concurrent with smoking status at the time. This may not be a fatal problem however as most people start smoking in their adolescence and given the addictiveness of the product, maybe smoking status at the time of the survey may indeed reflect smoking status at the time of death of the children. Quantity smoked however may indeed have changed. Maybe the death of a child altered the number of cigarettes smoked (it could in both directions).

Response 7: Thank you for your input. This is an excellent point and we have already added information concerning this issue in the limitation in the Discussion section.

Point 8: Indeed whilst the analysis by smoker status yields likely results, the analysis by number of cigarettes smoked provide results that appear contradictory. For example, paternal smoking 11-20 cigarettes yield ORs that are better than for 20+. Maybe if the ORs are presented to one decimal place rather than 2

Response 8: If the ORs are presented to one decimal place rather than 2, it is technically relevant; however it did not change the information provided in Table 4, with lower OR =  1.5 in 11-20 cigarettes/day compared to >20 cigarettes/day (OR = 1,6). Hence we presented the ORs to 2 decimals.

Point 9: There is not much information on how the death history is constructed. It may be useful to present in Table 3 the number of child deaths by smoking characteristic of parent. And what is the unit of analysis. Was it actual number of children born alive who died before age 5, and actual number of children born alive and who did not die in first five years of life cross tabulated by say paternal smoking status’ Or was it the number of families with a history of at least one under-five child death.”

Response 9: DHS’ respondents, which were the couple who live in the same household, were asked about the mother’s pregnancy history. We were looking at the result and then recoded actual number of children born alive who died before age five as “1” and other histories (actual number of children born alive and who did not die in first five years of life) as “0”. Final data reported was the number of couples with a history of at least one under-5 year’s child deaths.

Point 10: A review of the child death rates by country presented in table 2 is interesting. Indonesia, Cambodia and Lao are all statistically different from each other. Timor-Leste has rates that are consistent with Indonesia and Cambodia. The authors go on to collapse the data across the first three countries without any commentary.

Response 10: We apologize for not making it clear. We did not intend to collapse the data across the first three countries in Table 2; instead we presented the data separately for each country.

Point 11: Also why is it that Indonesia has the lowest mortality rate at the same time that it has the highest paternal smoking. Is it because they have the lowest maternal smoking rate is lower in that country.

Response 11: The possible explanation is that in Indonesia, most fathers are workers and they might choose to not smoke in the presence of their children. Besides, the maternal and child health services (monitoring child health status and nutrition) and programs (compulsory immunization, nutrition education, exclusive breastfeeding, weaning food) have been improved, supported by the implementation of the National Health Insurance in 2014 which opened more extensive opportunities for more accessible healthcare in the Indonesian population.

Point 12: Discussion and conclusion

I have read the discussion and conclusion but there is no point reviewing these in detail until the authors consider the suggested actions presented above.

Response 12: We hope our revision meet your approval. Again, we thank the reviewer and appreciate the opportunity to revise our work for consideration for publication in IJERPH.

Round 2

Reviewer 1 Report

No more comments.
